# Auxin-Producing Bacteria from Duckweeds Have Different Colonization Patterns and Effects on Plant Morphology

**DOI:** 10.3390/plants11060721

**Published:** 2022-03-08

**Authors:** Sarah Gilbert, Alexander Poulev, William Chrisler, Kenneth Acosta, Galya Orr, Sarah Lebeis, Eric Lam

**Affiliations:** 1Department of Plant Biology, Rutgers University, New Brunswick, NJ 08901, USA; sarahg19@email.unc.edu (S.G.); apoulev@sebs.rutgers.edu (A.P.); ikennethacosta@gmail.com (K.A.); 2Environmental Molecular Sciences Laboratory, Pacific Northwest National Laboratory, Richland, WA 99352, USA; william.chrisler@pnnl.gov (W.C.); galya.orr@pnnl.gov (G.O.); 3Department of Microbiology and Molecular Genetics, Michigan State University, East Lansing, MI 48824, USA; lebeis.sarah@gmail.com

**Keywords:** duckweed-associated bacteria, *Microbacterium*, *Azospirillum*, auxin, *AXR1*, Arabidopsis

## Abstract

The role of auxin in plant–microbe interaction has primarily been studied using indole-3-acetic acid (IAA)-producing pathogenic or plant-growth-promoting bacteria. However, the IAA biosynthesis pathway in bacteria involves indole-related compounds (IRCs) and intermediates with less known functions. Here, we seek to understand changes in plant response to multiple plant-associated bacteria taxa and strains that differ in their ability to produce IRCs. We had previously studied 47 bacterial strains isolated from several duckweed species and determined that 79% of these strains produced IRCs in culture, such as IAA, indole lactic acid (ILA), and indole. Using *Arabidopsis thaliana* as our model plant with excellent genetic tools, we performed binary association assays on a subset of these strains to evaluate morphological responses in the plant host and the mode of bacterial colonization. Of the 21 tested strains, only four high-quantity IAA-producing *Microbacterium* strains caused an auxin root phenotype. Compared to the commonly used colorimetric Salkowski assay, auxin concentration determined by LC–MS was a superior indicator of a bacteria’s ability to cause an auxin root phenotype. Studies with the auxin response mutant *axr1-3* provided further genetic support for the role of auxin signaling in mediating the root morphology response to IAA-producing bacteria strains. Interestingly, our microscopy results also revealed new evidence for the role of the conserved *AXR1* gene in endophytic colonization of IAA-producing *Azospirillum baldaniorum* Sp245 via the guard cells.

## 1. Introduction

The phytohormone indole-3-acetic acid (IAA) is the most commonly occurring auxin found in nature and is produced by both plants and bacteria through a similar biosynthetic pathway [1,2,3]. In addition to its role in gravitropism and cell elongation, IAA can alter plant root architecture to increase the efficiency of nutrient acquisition or its action may be downregulated by the plant to optimize defense against pathogens [4,5,6,7]. Homeostasis of auxin activities through biosynthesis, conjugation, oxidation, and transport is important for plants to maintain a balance between defense response and growth [8]. According to the “cry for help” hypothesis, when a plant detects a pathogen, it alters its root exudation profile to recruit and assemble a beneficial microbiome [9,10]. A recent study showed that elevated reactive oxygen species (ROS) levels in Arabidopsis activated IAA production by *Bacillus velezensis* FZB42, which is necessary for its colonization [11]. Microbes may modulate plant defense or growth by manipulation of the auxin pathway in the host by directly producing IAA themselves or altering levels of endogenous IAA levels through effects on plant auxin synthesis and/or conjugation pathways [1,12,13,14,15,16,17,18,19].

With access to a large collection of aquatic plants in the duckweed family, we previously carried out an initial survey of auxin-producing bacteria within the duckweed microbiome of diverse species and genera [20]. In the bacterial supernatant from a subset of strains that were studied, a variety of IRCs were detected, such as IAA, indole-3-lactic acid, and indole. We sought to examine the specificity of active auxins by coculturing the strains with the model plant *Arabidopsis thaliana* (hereafter Arabidopsis). We chose to use Arabidopsis as the host plant in our studies because of the vast resource of characterized mutant lines and high-quality genomic resources available for this species. In this study, we used the auxin response mutant *axr1-3* for comparison to wild type as both backgrounds have a similar root phenotype grown under sterile conditions. The highly conserved *Auxin Resistant-1* (*AXR1*) gene is involved in downstream auxin signaling through canonical auxin sensor F-box proteins, such as the TIR1 (transport inhibitor response 1) receptor [21,22]. AXR1 acts via conjugation of the small, ubiquitin-related protein NEDD8 (also called RUB in plants and yeast) to CULLIN1-containing E3 ligase SCF^TIR1^ [23] and enhances their ubiquitylation of target IAA/Aux repressor proteins, such as AXR3, for rapid turnover via the proteasome [24]. The AXR1 protein is structurally related to a ubiquitin-activating E1 enzyme, and the *axr1-3* mutant shows reduced sensitivity to IAA in the roots along with various other tissues [25,26].

In addition to the lack of characterized mutants in auxin response, duckweeds reproduce asexually, with some species doubling themselves in one to two days, making it challenging to observe the effects of plant-growth-promoting bacteria on an individual plant [27]. Some duckweed species lack roots or have multiple roots, which also makes it difficult to observe a bacteria-induced auxin root phenotype [28]. Previously, we found that most of our isolates from bleach-treated duckweed were of the phyla Proteobacteria, Firmicutes, and Actinobacteria [20], which is similar to reports from model land plants, such as the dicot Arabidopsis [29,30,31,32]. In addition, epiphytic and endophytic *Azospirillum* strains of PGPB isolated from wheat behaved similarly when cocultivated with duckweed [33], suggesting likely conservation of plant–microbe association mechanisms. A more recent global survey of duckweed-associated bacteria (DAB) community structure provided robust support for the core microbiome of duckweeds being very similar to those found in leaf tissues of monocots and dicots, indicating a highly conserved mode of selection for many bacteria taxa across plant species [34].

As we had previously found that strongly associated bacteria from duckweed tissues produce different IRCs, we sought to determine whether these strains have plant-growth-promoting abilities. We hypothesized that strains producing different IRCs, such as indole-3-acetic acid, indole lactic acid, and indole, may have different colonization patterns and morphological effects on plants. In this work, 21 bacterial strains capable of producing IRCs were individually inoculated onto gnotobiotically grown Arabidopsis seedlings from wild type and the auxin response mutant line *axr1-3.* Plant morphological responses and the pattern of bacterial colonization were assessed to evaluate the effects of bacteria on root development in these genetic backgrounds.

## 2. Results

### 2.1. The Salkowski Assay Is Insufficient as a Proxy for Auxin Production by Bacteria

Upon addition of the Salkowski reagent to bacterial supernatant, a color change from yellow to red can indicate that an IRC, such as IAA, is present. With its simplicity and low cost, this method is commonly used in a high-throughput format to screen for bacteria capable of producing auxin, which refers to the well-known ability of IAA to produce a stereotypical root phenotype. Using the Salkowski assay, we previously screened a collection of 47 bacterial isolates from 16 duckweed ecotypes for their ability to produce IRCs in vitro [20]. These duckweed-associated bacteria (DABs) were classified as “pink-type” or “brown-type” depending on the color change of their supernatant when the Salkowski reagent was added [20]). Using a combination of synthetic standards for various IRCs and liquid chromatography–mass spectrometry (LC–MS), we determined that indole-3-acetic acid (IAA) results in a pink color change and indole results in a brown color change. Our work thus demonstrated that using optical density at a single wavelength (typically at 530–535 nm) with the Salkowski reagent, as is commonly done, would not be sufficient to accurately identify IAA-producing bacteria due to high occurrence of false positives. Through LC–MS, we demonstrated the production of indole lactic acid in addition to IAA from one strain of DAB, *Herbaspirillum* RU5E [20]. Our results show that more than one type of IRC can be produced by a single DAB and that indole producers can be commonly found among Salkowski-positive bacteria strains.

In this study, we tested whether strains that were able to produce IRCs in vitro as determined by the Salkowski assay were also able to alter the physiology of a host plant, such as the production of a short root phenotype when inoculated onto Arabidopsis seedlings. This phenotype is indicative of an auxin response that results in decreased primary root length while increasing lateral root number and root hairs [35]. As positive controls, we used *Azospirillum* strains originally isolated from wheat, Sp7 and Sp245, which are well-studied PGPBs that can produce IAA and affect growth in various plant species, including Arabidopsis [36]. Of the 21 screened IAA-producing and/or indole-producing DABs, only four IAA-producing strains caused a short root phenotype (Figure 1). The strains that inhibited primary root length—*Microbacterium* sp. RU1A, *Microbacterium* sp. RU1D, *Microbacterium* sp. RU19A, and *Microbacterium* sp. RU19B—were derived from the duckweed genus *Lemna.* Only one other bacterial strain of the 21 tested was of the genus *Microbacterium*, and this strain, *Microbacterium* sp. RU33B, which was isolated from a duckweed in the genus *Wolffia,* did not inhibit primary root length in Arabidopsis. Under brightfield microscopy, we observed that primary root length inhibition was accompanied by an increase in root hairs, as demonstrated in wild-type Arabidopsis roots cocultivated with RU1A (Appendix A). This phenotype is thus indicative of auxin response in the plant by the bacteria treatment.

None of the brown-type strains that produced a significant amount of indole and a small but detectable amount of IAA [20] caused a short root phenotype in Arabidopsis. Moreover, strains that turned the darkest shade of red by the Salkowski assay and were first suspected to be high producers of IRCs did not produce a short root phenotype (Figure 1). In our assays, exogenous tryptophan, a precursor for a common pathway of IAA biosynthesis in plants and bacteria, was not added to the bacteria growth medium before inoculation onto the plant. Exogenous tryptophan would thus need to be supplied by the plant if any was taken up by the bacterial strains. Of the five strains that tested positive in the Salkowski assay without exogenous L-tryptophan [20], only one strain, *Microbacterium* RU1D, caused a short root phenotype (Figure 1). Therefore, the ability to produce IRCs, including IAA, without exogenous L-tryptophan is insufficient for the bacteria to cause a short root phenotype.

### 2.2. Comparison of IAA Quantification Methods

We next asked whether the short root phenotype caused by *Microbacterium* strains may be quantitatively related to their ability to produce higher levels of IAA by the bacteria in vitro. LC–MS was used to identify and quantify the amount of free IAA in the supernatant of various bacterial strains that tested positive in our Salkowski assay. The molecular weight of free IAA is 175 g/mol, with positive ionization resulting in a molecular ion at a *m*/*z* value of 176 [M + H] and a fragment at *m*/*z* of 130, as previously determined [20]. The retention time of free IAA in our LC–MS system was determined to be approximately 9.7 min from our previous work [20]. A free IAA standard was used to determine the HPLC UV absorbance signal at 280 nm for quantification. The resulting standard curve equation was generated: y = 5722x − 193.47 with an R^2^ value of 1.00. Using three biological replicates of 1 µL injections each, we calculated the % recovery for free IAA in our extraction with 5 ng/µL spike samples. The free IAA spike in the LB medium was 2.408 ng/µL ± 0.173 ng/µL (48% recovery), and the amount of free IAA spike in the TSB medium was 2.750 ng/µL ± 0.184 ng/µL (55% recovery).

We screened the Salkowski-negative control strain *Bacillus* RU3D, two Salkowski-positive control strains *Azospirillum* Sp7 and Sp245, two *Microbacterium* strains RU1A and RU19A that caused a short root phenotype, *Microbacterium* RU33B that did not cause a short root phenotype, as well as four additional strains that do not produce a short root phenotype and yet were top producers of IRCs based on the Salkowski assay (RU5E, RU20A, RU33A, and RU37A). The strains incapable of causing a short root phenotype in Arabidopsis seedlings all produced lower than 1 ng/µL of free IAA (Figure 2). Positive control strain Sp245 produced a similar level of IAA as previously reported [37]. In sum, our comparative analysis across these 10 strains of plant-associated bacteria indicates a requirement of higher levels (>1 ng/μL in the culture media) of IAA production by the particular strain for their ability to alter root development in Arabidopsis. By comparing the Salkowski assay to LC–MS, we determined that LC–MS is clearly a more accurate method for predicting an auxin root phenotype as the former cannot resolve various IRCs, many of which do not function as auxins.

### 2.3. Inoculation of Bacteria on Auxin Response Mutant Plants

To further confirm that the short root phenotype we observed with the bacterial strains that can produce high levels of IAA is indeed mediated through the auxin response pathway, we tested a subset of bacteria on a characterized Arabidopsis auxin response mutant at the *AXR1* locus to determine whether their effect on root length would be suppressed. As expected, exogenously applied 1 µM IAA no longer inhibited root length in this mutant background in comparison to wild-type seedlings, thus verifying that the IAA-induced short root phenotype requires this known auxin response mediator (Figure 3). DAB RU1A also failed to inhibit root length in *axr1-3* seedlings (Figure 3). Similarly, the positive control IAA-producing strain Sp245 no longer inhibited root length in *axr1-3* (Figure 3). In summary, these results indicate that the *AXR1* gene is involved in the root response to IAA-producing bacteria, such as RU1A and Sp245. Coupled with the lack of any root response in the various strains of DABs tested, which showed little to no IAA production, our data supports the hypothesis that the auxin produced by these plant-associated bacteria, when produced at sufficiently high levels, can mediate the physiological changes in the roots of host plants via their phytohormone pathways.

### 2.4. Colonization of Auxin-Producing Bacteria on Wild-Type Plants

We also investigated how colonization patterns of IAA-producing bacteria on plant roots may vary considering their different abilities to inhibit root length. We compared negative control *Bacillus* strain RU3D, which does not produce detectable IAA, to strains that produce a short root phenotype (IAA-producing *Microbacterium* RU1A and *Azospirillum baldaniorum* Sp245) and low IAA-producing strains that do not produce a short root phenotype (*Microbacterium* RU33B and *Herbaspirillum* RU5E). After treatment of Arabidopsis seedlings with each of the bacteria separately for 7 days, we used high-resolution 3D confocal microscopy with nucleic acid binding dyes to observe localization of the bacteria on inoculated gnotobiotic plant tissues from these seedlings. While nuclear DNA is also stained by these dyes, the size and morphology of the stained bodies readily distinguish them from the stained bacteria colonies. RU1A and Sp245 were found to be more abundant on the root surface than RU33B and RU5E (Figure 4). Imaging leaf tissues revealed that RU33B is more abundant on the leaves than the roots (Appendix A). In contrast, our data indicated that RU1A associated more strongly with root than leaf tissues of Arabidopsis seedlings (Appendix A). The lack of a short root phenotype by RU33B may thus result from low production of IAA as well as less efficient bacterial attachment and epiphytic colonization on Arabidopsis tissues, especially roots.

### 2.5. Colonization of Auxin-Producing Bacteria on Auxin Response Mutant Plants

Although RU5E was not highly abundant on the root surface and did not cause a short root phenotype, we observed detectable colonization under the root epidermis (Figure 5). Similarly, RU1A and Sp245 could also colonize the intercellular space beneath the root epidermis, suggesting that these could be endophytic bacteria (Figure 5). The pattern of RU5E and RU1A colonization did not change in the roots of the auxin response mutant *axr1-3*; however, Sp245 became unable to colonize the root epidermis of *axr1-3* plants and was instead more abundant on the root surface (Figure 5). Interestingly, on wild-type leaf tissues, Sp245 appeared to often target and accumulate inside the open stomata, which are pores located on the leaf surface and used for gas exchange and water transpiration (Figure 6). Strikingly, the leaf surface of *axr1-3* mutants showed no targeting of Sp245 at the stomata and were more randomly aggregated at the intercellular grooves (Figure 6). This suggests a potential role of *AXR1* in mediating endophytic colonization of IAA-producing Sp245 by targeting the stomatal pore as a point of entry, perhaps via a guard-cell-specific signaling pathway.

## 3. Discussion

### 3.1. Limitations of the Salkowski Assay as a Screen for PGPB

To utilize auxin-producing bacteria for agricultural applications, such as with synthetic bacterial communities, it is important to elucidate the role and mechanism of auxin signaling in the context of the plant microbiome [1,5,17,38]. Out of the 21 DAB strains capable of producing IRCs in our previous study [20], we identified only four *Microbacterium* strains that caused a short root phenotype in Arabidopsis. While the *Microbacterium* RU33B strain produced a positive Salkowski assay result indicative for synthesis of IRCs, it did not cause a short root phenotype in Arabidopsis seedlings. Using LC–MS to accurately quantify IAA in a collection of plant-associated bacteria isolates, we found that high levels of IAA (>1 ng/μL) in the bacteria’s growth medium correlated with the strain’s ability to cause a short root phenotype in Arabidopsis. Strains that were top producers of IRCs based on the Salkowski assay results (but, in many cases, apparently did not correspond to IAA) were not able to cause a short root phenotype. Thus, this commonly used colorimetric assay for detecting auxin-producing strains can often result in false positives [39,40,41,42]. Kuźniar et al. [43] detected IAA and IAA conjugates from endophytic bacteria isolated from winter wheat species using a combination of the Salkowski assay and LC–MS. They further tested bioactivity of the bacterial supernatant on wheat coleoptile segments and found the conjugates had lower biological activity in comparison to IAA. Our results highlight the importance of using LC–MS in combination with the Salkowski assay to screen for PGPB across plant species and identify bona fide auxin-producing bacteria strains.

While the correlation between higher levels of IAA-producing capability in the bacteria strain and its ability to modify root development of Arabidopsis seedlings is striking in this study (compare Figure 1 and Figure 2), the sample number in terms of different genera and strains of bacteria tested is likely too low in our current dataset to make a general statement about the threshold of IAA production needed to be effective in planta. Further testing of additional plant-associated bacteria with varying capacity for IAA production would be necessary to build on this initial work to define the threshold level(s) of auxin biosynthesis by the bacteria and its ability to modify host root morphology. As a recent example, microbial-community-derived auxin was posited to play a possible role in increasing *Lemna minor* fitness as measured by the increased number of plants, although this work relied on only using the Salkowski assay to infer auxin production by the bacteria [44]. Similarly, *Bacillus safensis* strains were screened for IAA-producing capability solely using the Salkowski assay, and their function in inducing Cd stress tolerance and promoting plant growth was partly based on the strains’ ability to produce auxins [41,42]. Confirmation of this type of results by applying the more definitive LC–MS method to determine the identity and quantity of auxin(s) that are being produced would be important. Future work to quantify the concentration of DAB-derived IAA produced in vivo will be invaluable for creating synthetic DAB communities and understanding how they can be deployed to improve duckweed growth [45].

### 3.2. Colonization of Bacteria That Produce Different Indole-Containing Compounds

In this study, we found that DAB strains that caused a short root phenotype in Arabidopsis were abundant on the root surface. For example, *Microbacterium* RU1A appeared to be more abundant on the root than the leaf tissues. This contrasts with *Microbacterium* RU33B, which was more abundant on the leaf surface than on the root and did not cause a short root phenotype. In contrast to these *Microbacteria* isolates, *Herbaspirillum* RU5E produced a higher concentration of indole lactic acid than indole-3-acetic acid in vitro. Despite it being endophytic in the root, albeit at low abundance, RU5E did not cause a short root phenotype. This suggests that the duckweed microbiome can produce different indole-containing compounds at various concentrations, with strains having unique colonization patterns and potentially occupying different niches [46,47,48]. Whether these colonization patterns of different DAB strains may be altered in the presence of other microbes will need to be examined in future synthetic community studies to further define the rules governing the ecological interactions that give rise to the microbiome’s structure on host plants.

How plants select for beneficial bacteria while defending against pathogens is not yet well understood [49,50]. Over the past decade, the complex roles that guard cells can play in plant responses to biotic and abiotic stresses have been revealed [51]. In addition to the well-established function of regulating gas exchange and transpiration, these specialized pores also play critical roles in microbial defense through their regulation via the phytohormone abscisic acid as well as others, such as salicylic acid and jasmonic acid [52]. By regulating closure of the aperture between the guard cells, these phytohormones can control the physical barrier that often allow entry of microbes into the intercellular space of plant aerial tissues. Bacterial phytotoxin, such as coronatin, has been demonstrated to be an important virulence determinant through its ability to maintain the stomata in the open state, while common molecular patterns of bacteria, such as the flagellar peptide flg22 that induce basal immunity functions, are known to induce closure of the stomata. In this study, we used *Azospirillum baldaniorum* Sp245 as a positive control since it is a well-characterized IAA-producing endophyte. Including this control in our work led to the unexpected finding that the *AXR1* gene, which is known to be involved in downstream auxin signaling, is necessary for endophytic root colonization of Sp245. Our microscopy results also uncovered the potential role of the guard cells in mediating bacterial entry for this strain, as shown by the remarkable concentration of bacteria inside the open stomata of wild-type plants but not in the *axr1-3* mutant background. In contrast, endophytic colonization of *Microbacterium* RU1A and *Herbaspirillum* RU5E were not altered in the *axr1-3* mutant. These observations suggest that plants have multiple mechanisms to regulate endophytic colonization by different IAA-producing bacteria, one of which requires guard-cell-specific signaling in an AXR-1-dependent manner.

## 4. Materials and Methods

### 4.1. Bacterial Strains and Media

Bacterial strains were previously isolated from surface-sterilized duckweed ecotypes as described in Gilbert et al. [20] using either a salt/detergent solution alone or with a bleach wash. Well-characterized IAA-producing *Azospirillum* strains Sp7 [53] and Sp245 [54] isolated from wheat tissue were used as controls. Bacterial strains were stored at −80 °C in LB (Miller’s) from IBI Scientific (Dubuque, IA, USA) or tryptic soy broth (TSB) (Hardy Diagnostics, Springboro, OH, USA) depending on the medium of isolation, and supplemented with 40% (*v*/*v*) sterilized glycerol. To isolate single colonies, bacteria from a glycerol stock was spread onto an agar plate (LB or TSB depending on the medium of isolation) and then stored at 28 °C for 2 days or until single colonies were grown. Next, 6 mL liquid cultures of LB or TSB broth were made from a single colony and grown for 1 day at 28 °C and shaken at 240 rpm except for RU33B cultures, which were grown for 2 days at the same temperature and rpm due to slower growth. Bacteria 16S rRNA gene sequence data are available at NCBI GenBank under accession numbers MH217512–MH217560.

### 4.2. Colorimetric Detection of Indole-Related Compounds

For each strain, a single colony was used to inoculate 6 mL of liquid LB medium with 5 mM L-tryptophan. For DAB 33B, liquid TSB with 5 mM L-tryptophan was used instead due to difficulty growing on LB medium. After 48 h of growing at 28 °C with shaking at 240 rpm, 1 mL of culture was centrifuged for 5 min at 14,000× *g* rpm to collect the supernatant. The original Salkowski assay based on the Gordon and Weber protocol was adapted for a 96-well format [39]. In a Corning 96-well clear-bottom white plate, 100 μL of the supernatant was added to 200 μL of Salkowski reagent (10 mM FeCl_3_, 97% reagent grade, and 34.3% perchloric acid, ACS grade) in duplicate. After incubating samples with the Salkowski reagent at room temperature for 30 min, the color change was recorded. A BioTek Synergy HT microplate reader was used to determine the absorbance (O.D.) at a single wavelength of 530 nm. To estimate the amount of indole-related compounds at 530 nm, an IAA standard curve was generated by suspending IAA (Gibco Laboratories, Life Technologies, Inc., New York, NY, USA) in 100% acetonitrile at a concentration of 1 mg/mL and diluting in LB medium or TSB to a concentration of 100, 50, 20, 10, 5, and 0 μg/mL. Sterile LB medium with 5 mM L-tryptophan and sterile TSB with 5 mM L-tryptophan were used as controls. The concentration of IRCs at 530 nm of the sterile control sample, either LB or TSB depending on the bacterial medium used, was subtracted from the concentration of indole-related compounds at 530 nm of the bacterial samples to obtain a background-subtracted concentration.

### 4.3. Extraction of IAA

From glycerol stocks, bacterial strains were streaked onto an LB or TSA (for DAB 33B) agar plate and grown at 28 °C. A single colony was used to inoculate a starter culture of 6 mL liquid LB medium, supplemented with 5 mM L-tryptophan (Sigma-Aldrich, St. Louis, MO, USA), and grown at 28 °C and 240 rpm. After 24 h, the starter culture was used to make a 60 mL culture of liquid LB medium, supplemented with 5 mM L-tryptophan, at OD_600_ 0.01. The cultures were grown at 28 °C and 240 rpm for 24 h. The supernatant was collected at 8000× *g* at 4 °C. For IAA spike samples, 300 µg of IAA was added to the culture by first generating a 1 mg/mL IAA solution in 100% acetonitrile and diluting to 100 µg/mL IAA solution in LB medium, supplemented with 5 mM L-tryptophan. Samples were then acidified with 1N HCl to a pH of 3.0. The samples were then separated into 20 mL aliquots for biological triplicates.

A Sep-Pak C18 cartridge (360 mg sorbent, 55–105 µm particle size) was prepared for each sample by washing with 10 mL of 100% acetonitrile followed by 10 mL of water. The acidified supernatant was passed through the C18 cartridge. The C18 cartridge was then washed with 10 mL of water and eluted with 5 mL of 80% (*v/v*) acetonitrile. The eluate was centrifuged at 12,000× *g* rpm for 5 min at 4 °C to remove solid particles. A 20 ng/µL solution of IAA was suspended into 100% acetonitrile for use as a standard in mass spectrometry. Acetonitrile of HPLC grade and HCl of ACS grade were used for the experiment, and water was prepared from Millipore Synergy 185.

### 4.4. LC–MS

Samples were separated and analyzed by a UPLC/MS system with the Dionex^®^ UltiMate 3000 RSLC ultrahigh-pressure liquid chromatography system consisting of a workstation with ThermoFisher Scientific’s Xcalibur v. 4.0 software package combined with Dionex^®^’s SII LC control software, solvent rack/degasser SRD-3400, pulseless chromatography pump HPG-3400RS, autosampler WPS-3000RS, column compartment TCC-3000RS, and photodiode array detector DAD-3000RS. After the photodiode array detector, the eluent flow was guided to a Q Exactive Plus Orbitrap high-resolution high-mass-accuracy mass spectrometer (MS). Mass detection was full MS scan with low-energy collision-induced dissociation (CID) from 100 to 1000 *m*/*z* in positive ionization mode with electrospray (ESI) interface. Sheath gas flow rate was 30 arbitrary units, auxiliary gas flow rate was 7, and sweep gas flow rate was 1. The spray voltage was 3500 volts (−3500 for negative ESI) with a capillary temperature of 275 °C. The mass resolution was 140,000, and the isolation window was 4.0 mDa. Substances were separated on a Phenomenex^TM^ Kinetex C8 reverse-phase column, size 100 × 2 mm, particle size 2.6 mm, pore size 100 Å. The mobile phase consisted of two components: solvent A (0.5% ACS grade acetic acid in LCMS grade water, pH 3–3.5) and solvent B (100% acetonitrile, LCMS grade). The mobile phase flow was 0.20 mL/min, and a gradient mode was used for all analyses. The initial conditions of the gradient were 95% A and 5% B. After 30 min, the proportion reached 5% A and 95% B, which was kept for the next 8 min. During the following 4 min, the ratio was brought to initial conditions. An 8 min equilibration interval was included between subsequent injections. The average pump pressure using these parameters was typically around 3900 psi for the initial conditions.

Putative formulas of IAA metabolites were determined by performing isotope abundance analysis on the high-resolution mass spectral data with Xcalibur v. 4.0 software and reporting the best fitting empirical formula. Database searches were performed using reaxys.com (RELX Intellectual Properties SA, Neuchatel, Switzerland) and SciFinder (American Chemical Society, Washington, DC, USA). Using the external standard of IAA with concentrations of 2.5, 5, 50, and 100 ng/µL with 0.2 µL injections, we calculated the concentration of free IAA in the samples using the peak area in UV chromatograms at 280 nm. To calculate the concentration in the original culture, the concentration was then divided by four to account for the original culture volume being 20 µL and the final elution volume being 5 µL. The concentration of IAA in the LB or TSB medium control sample was then subtracted to obtain the final concentration of IAA produced by the bacteria.

### 4.5. Arabidopsis Growth Assay

The Arabidopsis growth assay was performed in a similar manner for observation of root lengths and microscopy. For each assay, 200 *Arabidopsis thaliana* (Col-0 ecotype) seeds (wild type or *axr1-3* genotype) were sterilized using 50% (*v*/*v*) bleach solution (0.3% sodium hypochlorite) in a 1.5 mL microcentrifuge tube for 4 min with continuous shaking using a vortex (Fisher Genie 2) shake setting of 6. The bleach solution was removed, and the seeds were washed four times in 1 mL of sterile water. After removing the water, the seeds were suspended in 0.1% (*w*/*v*) Difco agar granulated (Becton Dickinson, Sparks, MD, USA). Seeds were placed onto circular 100 × 15 mm plates containing 0.5× Murashige and Skoog (MS) modified basal medium with Gamborg vitamins (PhytoTech Laboratories, Lenexa, KS, USA), 1% sucrose, pH 5.7, 0.25% phytagel (Sigma-Aldrich, St. Louis, MO, USA). The seeds were vernalized at 4 °C in the dark for 2 days and then stored vertically in a growth chamber at 22 °C under 100 µmol m^−2^ s^−1^ of 12 h light. After 6 days, previously grown bacterial cultures were prepared by taking 1 mL of culture and centrifuging at 14,000× *g* rpm for 5 min. The supernatant was discarded, and the bacterial pellet was resuspended in sterile water to an OD_600_/cm of 0.7 (1.58 × 10^7^ CFU/mL was measured for RU1A). Bacterial cultures for heat-killed samples were autoclaved and centrifuged at 8000× *g* rpm for 5 min, and the pellet was diluted to an OD_600_/cm of 0.7 before plating 100 µL onto an LB plate to check vitality. Next, 100 µL of heat-killed or living bacterial solution was spread onto square 100 × 15 mm plates containing 0.5× MS, pH 5.7, 0.5% gellan gum powder (PhytoTech Laboratories, Lenexa, KS, USA). Media containing 1 µM IAA (Gibco Laboratories, Grand Island, NY, USA) was previously prepared by adding IAA dissolved in DMSO (Sigma-Aldrich, St. Louis, MO, USA) directly to the media before pouring and solidifying. Then, 6–12 seedlings (depending on the assay) were transferred onto each plate, which were then sealed with a self-adherent wrap (3M micropore surgical tape; Coban, St. Paul, MN, USA). Plates were then placed in the same growth chamber under the same conditions as previously described for 7 days until processing for all subsequent experiments. Pictures of plants were taken with a Nikon D5200 camera, and roots were measured using ImageJ. Water was prepared from Millipore Synergy 185 and sterilized using a 0.2 micron polyethersulfone syringe filter.

### 4.6. Confocal Microscopy

Five whole seedlings of sterile or bacteria-treated, from wild-type and *axr1-3* genotypes, were fixed in 1 mL of 4% paraformaldehyde overnight at room temperature. The solution was removed followed by washing twice with 1 mL of sterile phosphate buffer saline (1.37 M NaCl, 26 mM KCl, 10 mM Na_2_HPO_4_·7H_2_O, 17.6 mM KH_2_PO_4_, pH 7.4) and then storing at 4 °C. Images were acquired by EMSL (Richland, WA, USA) using a Zeiss LSM 710 scanning confocal microscope. The channels used were blue (calcofluor white), green (SYBR Gold DNA), red (chlorophyll autofluorescence), and gray (transmitted light).

## Figures and Tables

**Figure 1 plants-11-00721-f001:**
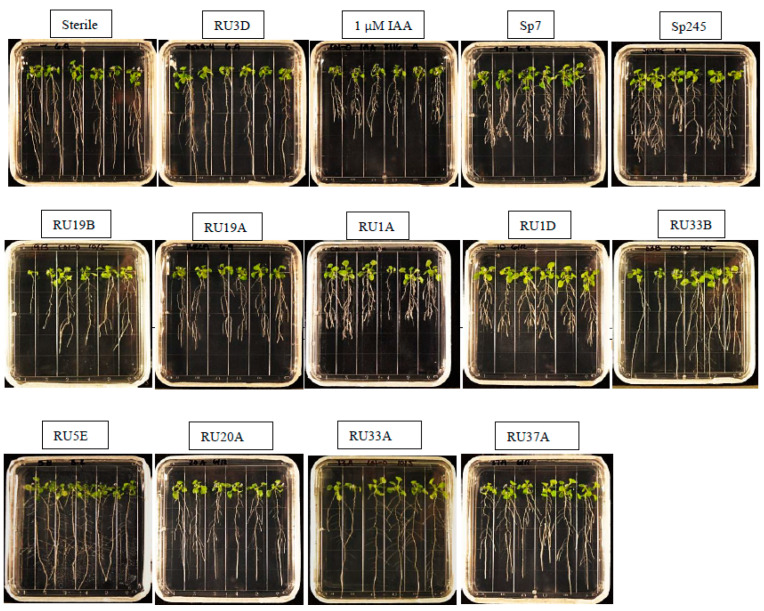
Bacterial strains that have a positive Salkowski assay result do not always result in primary root length inhibition. Representative image of wild-type root development after 7 days of treatment with bacteria or IAA. Out of 21 DAB strains producing a positive Salkowski assay result, only four *Microbacterium* strains—RU19B, RU19A, RU1A, and RU1D—caused an auxin root phenotype. *Azospirillum baldaniorum* strains Sp7 and Sp245 are known auxin-producing, plant-growth-promoting strains derived from wheat. *Bacillus* RU3D produced a negative Salkowski assay result and *Microbacterium* RU33B, *Rhizobium* RU20A, *Rhizobium* RU33A, *Herbaspirillum* RU5E, and *Azospirillum* RU37A produced a positive Salkowski assay result yet did not cause an auxin root phenotype.

**Figure 2 plants-11-00721-f002:**
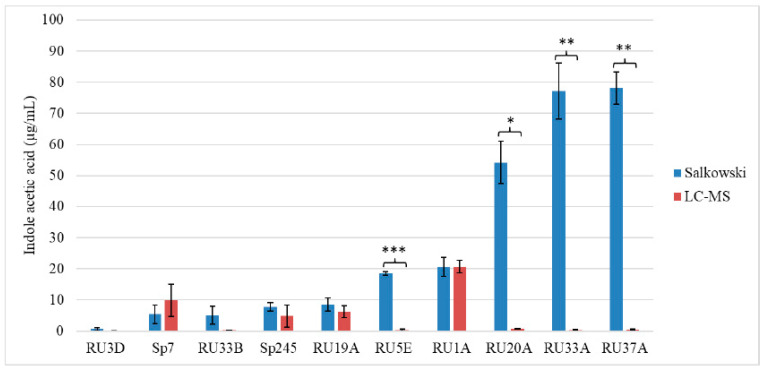
LC–MS and the Salkowski colorimetric assay quantify different concentrations of bacterial-derived indole acetic acid. Bars represent background subtracted mean values and standard deviation. Student’s T-test (* = *p* < 0.05, ** = *p* < 0.005, *** = *p* < 0.0005) was performed with n = 3.

**Figure 3 plants-11-00721-f003:**
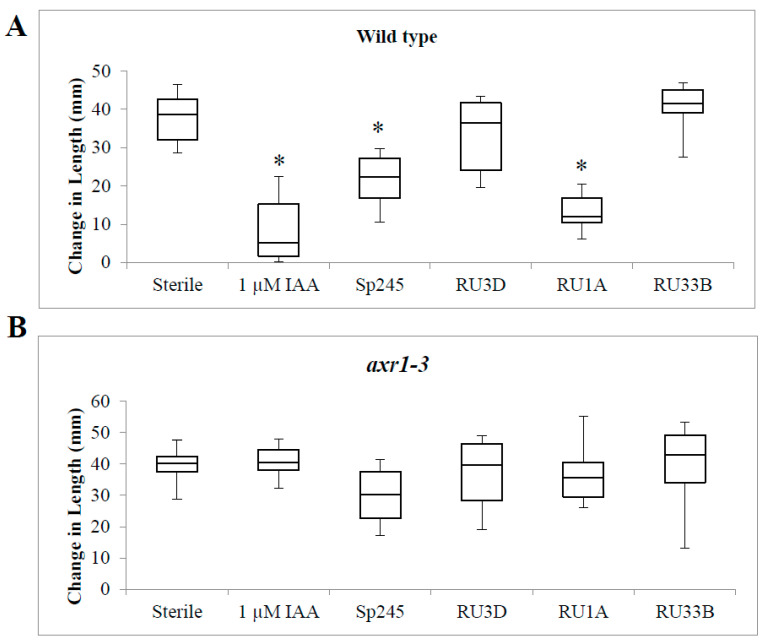
Auxin response gene *AXR1* is necessary for primary root length inhibition by IAA-producing strains *Azospirillum* Sp245 and *Microbacterium* RU1A. Change in primary root length after 7 days in the genetic backgrounds: (**A**) wild type and (**B**) auxin response mutant *axr1-3.* For box plots, horizontal lines represent the median, with the box representing the 25th and 75th percentiles, and the whiskers representing the minimum and maximum. For positive control, 1 μM IAA was used. Student’s T-test (*p* < 0.05) was performed (n = 18), and an asterisk indicates significant difference compared to the sterile control.

**Figure 4 plants-11-00721-f004:**
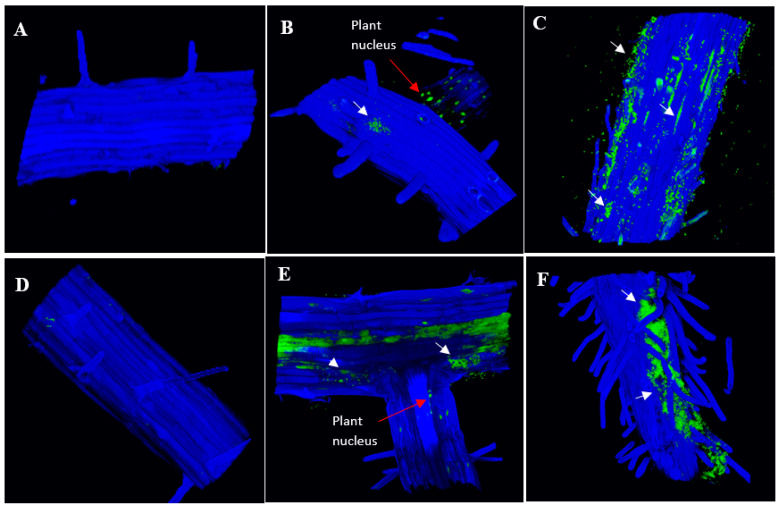
IAA-producing strains that inhibit root length are abundant on wild-type root tissue. A 3D reconstruction from confocal microscopy of wild-type Arabidopsis roots upon treatment with (**A**) no bacteria, (**B**) non-IAA-producing *Bacillus* RU3D, (**C**) IAA-producing *Microbacterium* RU1A, (**D**) IAA-producing *Microbacterium* RU33B, (**E**) IAA-producing *Herbaspirillum* RU5E, and (**F**) IAA-producing *Azospirillum baldaniorum* Sp245. The microscopy channels are blue (calcofluor white used to stain the cell wall) and green (SYBR Gold DNA used to stain the nuclei and bacteria). White arrows indicate bacteria location based on the size of the DNA-stained spots. Bacteria are shown as green spots that are smaller in size to plant nuclei.

**Figure 5 plants-11-00721-f005:**
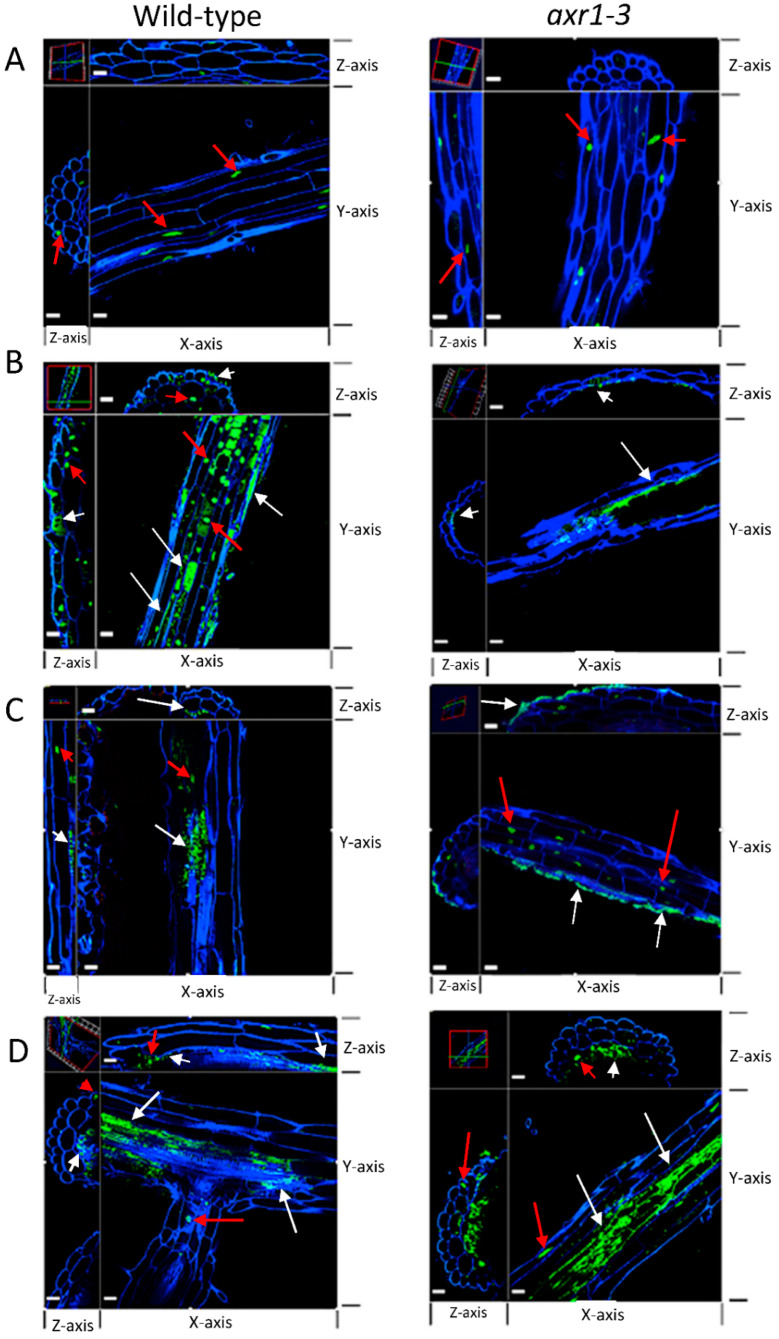
IAA-producing strains that inhibit root length differentially colonize *axr1-3* root tissue. Confocal microscopy showing orthogonal view of wild-type (left panels) and *axr1-3* (right panels) roots upon treatment with (**A**) no bacteria, (**B**) IAA-producing *Microbacterium* RU1A, (**C**) IAA-producing *Azospirillum baldaniorum* Sp245, and (**D**) low IAA-producing *Herbaspirillum* RU5E that does not inhibit root length. The microscopy channels for each dye are blue (calcofluor white used to stain cell wall) and green (SYBR Gold DNA used to stain the nucleus and bacteria). White arrows indicate bacteria locations based on the size and morphology of the DNA-stained spots. Bacteria are shown as green spots that are smaller in size to plant nuclei (shown with red arrows) and tend to form clusters. The size bar in white represents 20 μm on each panel. The 3D images are rotated at the *z*-axis at two different locations of the tissue shown (top and left sections of each panel as shown by the cross-hair in the upper left corner image) to illustrate transverse views at the location of the stained spots and demonstrate either epiphytic or endophytic locations.

**Figure 6 plants-11-00721-f006:**
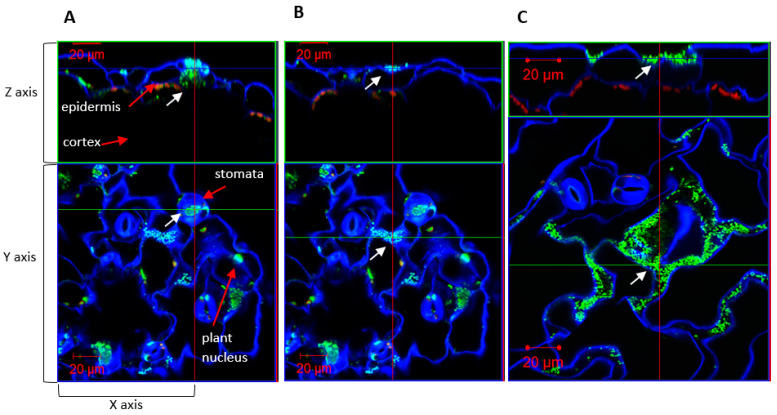
IAA-producing strain *Azospirillum baldaniorum* Sp245 colonizes leaf tissue through stomata. Confocal microscopy showing orthogonal view of (**A**,**B**) wild-type and (**C**) *axr1-3* leaf tissue inoculated with *Azospirillum baldaniorum* Sp245. Note that panels A and B present the *z*-axis inside the stomata and at the cell surface, respectively, for comparison. The microscopy channels are blue (calcofluor white), green (SYBR Gold DNA), and red (chlorophyll autofluorescence). White arrows indicate bacteria location based on the size of the DNA-stained spots. Bacteria are shown as green spots that are smaller in size to plant nuclei.

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
