# Peer review of "Auxin-Producing Bacteria from Duckweeds Have Different Colonization Patterns and Effects on Plant Morphology"

_plants, 2022, doi:10.3390/plants11060721_

Round 1
Reviewer 1 Report
The manuscript plants-1537934 reports an interesting topic. The manuscript was provided without any format and line number therefore I am giving my general perception of the manuscript and the data it reports. The authors reported in depth analysis of IAA producing microbes and further supported by spectroscopic and microscopic analysis which is a plus to the novelty. I include my comments, most of them are suggestions to improve the overall quality of the manuscript.
The abstract should report a concise overview of the manuscript. There should be a quantitative approach to mention the results.
AXR1 as a keyword seems awkward
The second paragraph of introduction is redundant and can be deleted or moved to methodology part.
Replace PGPBs with PGPB. Bacteria are already plural.
The repetitive phrases should be deleted throughout the manuscript.
I encourage the authors to cite recent references. some of them are provided herewith at the end.
It is not clear how the authors isolated the microbes? are they rhizospheric? endophytic?
The presentation of data needs to be improved substantially. It is not acceptable now.
The figure captions should provide a standalone information. They should be self-explanatory.
The results can be shortened to most significant ones. Delete the general statements.
The whole discussion section must be revised.
There should be a proper analysis of the findings obtained and their interpretation with previous works. Rendering your own previous works does not make any sense.
All of the observed variation cannot be correlated with IAA only.
LC-MS data is the most important confirmation to your data. This should be in a rigorous manner in the discussion.
A separate conclusion part must be added in the revised manuscript. I guess this ais a first written draft by the authors and must be improved substantially.
Some refernces for authors
- The good, the bad, and the ugly of rhizosphere microbiome. InProbiotics and plant health 2017 (pp. 253-290). Springer, Singapore.
- Application potentials of plant growth promoting rhizobacteria and fungi as an alternative to conventional weed control methods. Sustainable Crop Production.
- A Review on Practical Application and Potentials of Phytohormone-Producing Plant Growth-Promoting Rhizobacteria for Inducing Heavy Metal Tolerance in Crops. Sustainability, 12(21), p.9056.
- Insights into the Interactions among Roots, Rhizosphere, and Rhizobacteria for Improving Plant Growth and Tolerance to Abiotic Stresses: A Review. Cells, 10(6), p.1551.
- Efficacy of Indole Acetic Acid and Exopolysaccharides-Producing Bacillus safensis Strain FN13 for Inducing Cd-Stress Tolerance and Plant Growth Promotion in Brassica juncea (L.). Applied Sciences, 11(9), p.4160.
- Rhizosphere Bacteria in Plant Growth Promotion, Biocontrol, and Bioremediation of Contaminated Sites: A Comprehensive Review of Effects and Mechanisms." International Journal of Molecular Sciences22, no. 19 (2021): 10529.
Author Response
Here is our point by point response (highlighted in yellow)to each of this reviewer's critiques:
The abstract should report a concise overview of the manuscript. There should be a quantitative approach to mention the results. The abstract did provide a concise overview of the intro, methods, results, and conclusion. We did not itemize all the concentrations and root length differences since these are presented in the Results section and we do not want to be redundant.
AXR1 as a keyword seems awkward. We found that AXR1 (which stands for the highly conserved gene Auxin Resistant 1, originally identified in Arabidopsis, plays a role in the endophytic colonization of Sp245. This is a piece of novel information for readers studying the role of AXR1.
The second paragraph of introduction is redundant and can be deleted or moved to methodology part. We have removed redundancy in the second paragraph of the introduction.
Replace PGPBs with PGPB. Bacteria are already plural. This has been corrected in instances when multiple bacteria strains are discussed.
The repetitive phrases should be deleted throughout the manuscript. This has been corrected.
I encourage the authors to cite recent references. some of them are provided herewith at the end. Recent references related to duckweed microbiome studies and bacterial derived auxin are cited. A majority of the suggested references appear to come from a particular laboratory and are various review articles or book chapters. Since we have already cited more than 10 review articles in the reference list, we decline to include more which would be redundant in terms of relevance. We did include two additional references, one of which a primary research article suggested by this reviewer with a related paper that exemplified some of the past and continual problems in this field which rely solely on the Salkowski assay for auxin determination and inference.
It is not clear how the authors isolated the microbes? are they rhizospheric? endophytic? This information has been added and was cited in an earlier paper from our lab.
The presentation of data needs to be improved substantially. It is not acceptable now. This comment does not mention what specifically is not acceptable and needs improvement.
The figure captions should provide a standalone information. They should be self-explanatory. Figure captions are self-explanatory. This comment does not give specific examples of what is unclear and what information is lacking. Obviously many of the methods are described in the Methods section and not repeated in the legends.
The results can be shortened to most significant ones. Delete the general statements. Some general statements were provided to explain the context for the results. Which results need to be shortened or deleted? Specific examples of this type of criticisms would be very helpful.
The whole discussion section must be revised. This comment does not mention how or why the discussion must be revised.
There should be a proper analysis of the findings obtained and their interpretation with previous works. Rendering your own previous works does not make any sense. The results section mentions our previous work as a reference. This was done to provide relevant background information in order for the readers to understand our results which expands on our previous findings. Again, it is not clear what “previous works” does not make sense. It is not clear to us what “rendering” is referring to in this instance.
All of the observed variation cannot be correlated with IAA only. Based on our LC-MS analysis, we show that our bacteria are producing multiple indole related compounds. Thus, we tested the bacteria for their ability to cause a root phenotype and found that only strains producing high levels of IAA caused a short primary root and increased root hairs. Although we cannot exclude the possibility that the bacteria are producing other plant growth promoting hormones, we use the IAA related axr1-3 mutant to show the observed phenotypes are related to the AXR1 gene, a well known regulator of auxin responses.
LC-MS data is the most important confirmation to your data. This should be in a rigorous manner in the discussion. The Discussion states that our LC-MS data confirms our conclusion that the Salkowski assay has limitations for detecting PGPB. The methods for this technique were rigorosely described in the methods section, with detailed description of our quantification techniques and validation with pure standards. We do not see any point to include these methods description in the Discussion
A separate conclusion part must be added in the revised manuscript. I guess this ais a first written draft by the authors and must be improved substantially. Instructions of this Journal we found online say a separate conclusion section is optional. We have also checked other recently publish papers in this Journal and they do not have a separate section for Conclusions.
Reviewer 2 Report
The authors have conducted a number of experiments and the results are worthy for publication. However, the manuscript may lead readers to wrong direction because they grasp the results from one side and states too much. Figs. 1 and 2 show the results of only inconsistently selected species. Moreover, stating the conclusion as the title text of the figures is too rush and even arrogant.
IAA should not be the only bacterial metabolites that affect plant root growth and morphology. A pitfall of this paper is that the authors do not consider the combined effects of bacterial metabolites. For example, promotion of root hair formation was not observed by inoculation of simple IAA and strains RU19A, RU19B, but with certain bacteria such as RU1A. In Fig. 3, unlike the IAA compound, all bacterial inoculations gave similar tendencies to the root lengths of both wild-type and axr1-3 strains. This observation may attribute to a route unrelated to the IAA signaling pathway. It is also expected that productivity of IAA in the culture media for bacteria and plant varies widely or even reverse depending on the bacterium strain.
It seems appropriate to conclude some like that, of the twenty-one IAA producing bacteria, Microbacterium strains affected to the root growth and morphology similarly to the previously examined genus Azospirillum although their root colonization is different.
Additional comment:
The authors have skills to identify IAA and related compounds by LC-MS technology. It would provide further useful information for the reader to identify the causative substance of the abnormal color development. I am not sure but a table of IAA and related substance productivity of all the twenty-one bacteria compared with their colorimetry results maybe informative.
Author Response
Here is our point-by-point response (highlighted in yellow) to this reviewer's comments:
The authors have conducted a number of experiments and the results are worthy for publication. However, the manuscript may lead readers to wrong direction because they grasp the results from one side and states too much. Figs. 1 and 2 show the results of only inconsistently selected species. Moreover, stating the conclusion as the title text of the figures is too rush and even arrogant.
We are not sure why the manuscript leads the reader to the wrong direction and states too much. This referee needs to explain what the wrong direction is. Fig 1 shows the plant root phenotype when treated with bacteria that produced a positive Salkowski result. Fig 2 shows LC-MS confirmation of the Salkowski result for these bacteria. Many papers state the conclusion as the title text of the figures to make it clear to the reader what we intended to test for in the experiment. “Arrogant” seems quite an over-sensitive reaction to a common practice.
IAA should not be the only bacterial metabolites that affect plant root growth and morphology. A pitfall of this paper is that the authors do not consider the combined effects of bacterial metabolites. For example, promotion of root hair formation was not observed by inoculation of simple IAA and strains RU19A, RU19B, but with certain bacteria such as RU1A. In Fig. 3, unlike the IAA compound, all bacterial inoculations gave similar tendencies to the root lengths of both wild-type and axr1-3 strains. This observation may attribute to a route unrelated to the IAA signaling pathway. It is also expected that productivity of IAA in the culture media for bacteria and plant varies widely or even reverse depending on the bacterium strain.
Based on our LC-MS analysis, we show that our bacteria are producing multiple indole related compounds. Thus, we tested the bacteria for their ability to cause a root phenotype and found that only strains producing high levels of IAA caused a short primary root and increased root hairs in wild-type plants. This referee mis-stated that all bacteria gave similar effects to both wild-type and axr1 mutant plants! Although we cannot exclude the possibility that the bacteria are producing other plant growth promoting hormones or metabolites, we use the IAA related axr1-3 mutant to show the observed phenotypes requires a functioning AXR1 gene in the host plant. In this case, then all the bacteria tested do not show effects on root lenght anymore irrespective of their IAA production levels.
As for the root hair formation data shown in the supplemental data, simple addition of IAA does cause the same result which is why it is part of the auxin root phenotype. This is a well known effect of IAA and was stated in the discussion of the Results when the Suppl. Figure in question was cited.
We do not think this referee read our paper very carefully and perhaps that’s why he/she is dissatisfied with the presentation.
It seems appropriate to conclude some like that, of the twenty-one IAA producing bacteria, Microbacterium strains affected to the root growth and morphology similarly to the previously examined genus Azospirillum although their root colonization is different.
Yes, some of the Microbacterium strains affected root morphology similarly to the previously examined Azospirillum strains although their root colonization is different.
Additional comment:
The authors have skills to identify IAA and related compounds by LC-MS technology. It would provide further useful information for the reader to identify the causative substance of the abnormal color development. I am not sure but a table of IAA and related substance productivity of all the twenty-one bacteria compared with their colorimetry results maybe informative.
Line 103-106 states “Using a combination of synthetic standard of various indole-related compounds and liquid chromatography mass spectrometry (LC-MS), we determined that indole-3-acetic acid (IAA) results in a pink color change and indole results in a brown color change.” A table of the bacteria strains and their Salkowski result are referenced in Gilbert et al 2018, reference #20.
Reviewer 3 Report
Please see notes in the attached file.
1. I have some doubts about the Manuscript title, it does not reflect the content.
2. Please take into account the lack of Frame A for the Figure 5.
3. There is also a common mistake in the writing Arabidopsis (see comment in the article notes)
4. There are some gaps in describing methods.
5. Please, do not use the term auxins for the substances, determined by Salkowsky method. Appropriate term - indole-related compounds.

Author Response
Here we provided a point by point response (highlighted in yellow) to this reviewer's comments:
The interest of the present work is rather limited. The research described is extremely simple, giving the impression that the authors have tried to take advantage of already published results by incorporating minimal analyses, aimed at confirming already known facts. Thus, an important part of the results focus on re-describing what was collected in the article by Gilbert et al. (2018) (hence the high number of times that article is cited), while the Discussion repeats again what was referred to in Results, incorporating some theoretical truisms on plant-bacteria associations.
The first paragraph of this reviewer describes subjective criticisms with vague references. We do not know what specific issue this referee has except for the numerous citations of our previous work – which was the basis of our present study. In this present work, we have carried out much more careful quantitative comparison of IAA levels between multiple DABs and control strains. In addition, we deployed an Arabidopsis auxin response mutant to provide genetic proof that the short root phenotype observed with bacteria inoculation does require known components of plant auxin signaling. Finally, we carried out high resolution fluorescence microscopy to compare modes of bacteria colonization on leaf and root tissue of the inoculated plants. We do not agree with this referee that we “have tried to take advantage of already published results by incorporating minimal analyses, aimed at confirming already known facts”.
On the other hand, the bibliography needs a major revision. First of all, the author's instructions are not followed, both in the text (numerical citations should appear) and in the References section (editorial recommendations are not followed at any time). Secondly, there are errors in some of the citations in terms of the years of publication depending on whether they appear in the text or in the list. Finally, there are missing citations in the text that appear in the list, and citations that are in the text but not in the list.
The references and bibliography have now been correctly formatted and a few errors are corrected as noted by this referee. Thanks for pointing those out.
Reviewer 4 Report
The interest of the present work is rather limited. The research described is extremely simple, giving the impression that the authors have tried to take advantage of already published results by incorporating minimal analyses, aimed at confirming already known facts. Thus, an important part of the results focus on re-describing what was collected in the article by Gilbert et al. (2018) (hence the high number of times that article is cited), while the Discussion repeats again what was referred to in Results, incorporating some theoretical truisms on plant-bacteria associations.
On the other hand, the bibliography needs a major revision. First of all, the author's instructions are not followed, both in the text (numerical citations should appear) and in the References section (editorial recommendations are not followed at any time). Secondly, there are errors in some of the citations in terms of the years of publication depending on whether they appear in the text or in the list. Finally, there are missing citations in the text that appear in the list, and citations that are in the text but not in the list.
Author Response
Here we provide our point by point response (highlighted in yellow) to the coments of this referee (Rev. 3):
- I have some doubts about the Manuscript title, it does not reflect the content. What does the reviewer suggest would better reflect the content? Please be more specific so that we can address the issues or “doubts”.
- Please take into account the lack of Frame A for the Figure 5. What does the reviewer mean by “lack of Frame A”?
- There is also a common mistake in the writing Arabidopsis (see comment in the article notes) This has been corrected, thank you.
- There are some gaps in describing methods. Which specific methods have gaps?
- Please, do not use the term auxins for the substances, determined by Salkowsky method. Appropriate term - indole-related compounds. We used the term auxin to describe a compound that produces a root phenotype in the plant. We used the term indole-related compounds (abbreviated to IRCs) to describe compounds which may not necessarily produce a root phenotype in the plant but contain an indole group in their structure.
Reviewer 5 Report
A very interesting article.
Congratulations and I wish you further interesting subjects.

Author Response
No criticisms.
Round 2
Reviewer 1 Report
Dear Authors,
About 95% of the comments from the previous revisions have not been considered by you. A substantial revision taking into account the previous suggestion is therefore required. The discussion, the results, and introductory phrases must be improved. When you ask for specific comments these show your little interest to read the literature and improve your manuscript further. I am astonished how the handling editor can send the manuscript for peer review without having a conclusion of what is done in the MS.
Reviewer 3 Report
no comments
Reviewer 4 Report
The authors have corrected everything concerning the bibliography. However, the main objection expressed in the first review continues to discourage publication of the present work. Some slight modifications have been made to the text, but the substance of the study has not changed. On the other hand, it is difficult to improve this aspect, since this would require new trials to provide additional data and support the multiple hypotheses put forward, all of which are supported by data already published by the authors in previous articles or associated with well-established events.